# Immunotherapy and Immunotherapy Combinations in Metastatic Castration-Resistant Prostate Cancer

**DOI:** 10.3390/cancers13020334

**Published:** 2021-01-18

**Authors:** Dhruv Bansal, Melissa A. Reimers, Eric M. Knoche, Russell K. Pachynski

**Affiliations:** Division of Medical Oncology, Department of Internal Medicine, Washington University School of Medicine, St. Louis, MO 63110, USA; dbansal@wustl.edu (D.B.); mreimers@wustl.edu (M.A.R.); eknoche@wustl.edu (E.M.K.)

**Keywords:** metastatic castrate-resistant prostate cancer, immunotherapy, combination immunotherapy, cancer vaccines

## Abstract

**Simple Summary:**

Immunotherapy has changed the treatment paradigm of numerous malignancies such as non-small cell lung cancer and melanoma. To date, there has been only modest demonstrable efficacy of immunotherapy for prostate cancer. This lack of efficacy is likely due to the immunosuppressive tumor microenvironment. When we consider the fact that metastatic castrate-resistant state is the most lethal form of prostate cancer, there is an unmet need to increase the efficacy of immune therapies for this disease. The treatment paradigm has now shifted towards combinatorial regimens to enhance the anti-tumor immune response. These combinations with immunomodulatory agents in ongoing clinical trials include conventional agents such as chemotherapy and numerous novel agents. This review summarizes the clinical trials recruiting patients with metastatic castrate-resistant prostate cancer utilizing immunotherapeutic approaches.

**Abstract:**

Although most prostate cancers are localized, and the majority are curable, recurrences occur in approximately 35% of men. Among patients with prostate-specific antigen (PSA) recurrence and PSA doubling time (PSADT) less than 15 months after radical prostatectomy, prostate cancer accounted for approximately 90% of the deaths by 15 years after recurrence. An immunosuppressive tumor microenvironment (TME) and impaired cellular immunity are likely largely responsible for the limited utility of checkpoint inhibitors (CPIs) in advanced prostate cancer compared with other tumor types. Thus, for immunologically “cold” malignancies such as prostate cancer, clinical trial development has pivoted towards novel approaches to enhance immune responses. Numerous clinical trials are currently evaluating combination immunomodulatory strategies incorporating vaccine-based therapies, checkpoint inhibitors, and chimeric antigen receptor (CAR) T cells. Other trials evaluate the efficacy and safety of these immunomodulatory agents’ combinations with standard approaches such as androgen deprivation therapy (ADT), taxane-based chemotherapy, radiotherapy, and targeted therapies such as tyrosine kinase inhibitors (TKI) and poly ADP ribose polymerase (PARP) inhibitors. Here, we will review promising immunotherapies in development and ongoing trials for metastatic castration-resistant prostate cancer (mCRPC). These novel trials will build on past experiences and promise to usher a new era to treat patients with mCRPC.

## 1. Introduction

Per 2019 SEER estimates, prostate cancer comprises approximately 10% of all new cancer diagnoses, with over 98% of the patients alive at 5 years [1]. Recurrence after localized treatment occurs in about 1/3rd of the men, and these patients with recurrent disease eventually develop malignant cells resistant to androgen ablation alone [2,3]. This statistic points out that, while patients with prostate cancer have a low mortality, those with advanced prostate cancer eventually progress to the castrate-resistant disease [4]. Moreover, patients with a shorter prostate-specific antigen doubling time (PSADT) experience increased prostate cancer-specific and all-cause mortality [5].

While androgen deprivation therapy (ADT) is not curative, it does lead to an overall survival (OS) benefit of approximately 30 months in patients with metastatic disease [6]. Androgen deprivation can be achieved by surgical orchiectomy or medical castration using gonadotropin-releasing hormone receptor (GnRH-R) agonists or antagonists [7,8]. Currently, for patients with metastatic hormone-sensitive prostate cancer (mHSPC), additional first-line agents are frequently used in conjunction with an ADT backbone. These include three oral androgen receptor (AR)-targeted drugs—abiraterone acetate, apalutamide, and enzalutamide, as well as docetaxel chemotherapy [9,10,11,12,13,14,15,16].

The addition of these agents for patients with mHSPC has improved patient outcomes [17]. Based on the STAMPEDE clinical trial results, the 3-year failure-free survival (FFS)—defined as radiologic, clinical, or PSA progression or death from prostate cancer, was 75% in patients with mHSPC, treated with a combination of abiraterone and ADT [10]. The CHAARTED clinical trial showed that for patients with mHSPC treated with a combination of ADT and docetaxel, the median time to castrate resistance was 20.2 months [18]. In the phase III ARCHES clinical trial, at a median follow-up of approximately fourteen months, over 70% of the patients had developed castrate resistance while on enzalutamide [15]. Similar outcomes have been observed with apalutamide [14].

## 2. Background

Sipuleucel-T was the first therapeutic vaccine to be approved by the United States Food and Drug Administration (FDA) for patients with metastatic castration-resistant prostate cancer (mCRPC) based on the pivotal phase III IMPACT trial [19], and the first autologous cellular therapeutic vaccine for any cancer. While checkpoint inhibitors (CPI) immunotherapy has vastly improved the outcomes of patients with malignancies such as melanoma and non-small cell lung cancer, its efficacy in patients with prostate cancer to date has been modest [20,21,22].

There is evidence that a combination of CPIs induces clonal T cell expansion in patients with metastatic melanoma compared to those with mCRPC [23]. Exome sequencing of patients with prostate cancer revealed a low tumor mutation burden (TMB) even in heavily pretreated castration-resistant prostate cancer (CRPC) [24]. The low TMB, when compared to other malignancies, might explain the low immunogenicity of prostate cancer due to a smaller pool of neoantigens [25,26,27].

One of the major contributors to the poor response to CPIs is chronic inflammation leading to an immunosuppressive tumor microenvironment (TME) [22,28]. Prostate cancer cells express numerous tumor-associated antigens (TAA), such as PSA, prostate-specific membrane antigen (PSMA), prostatic acid phosphatase (PAP), and prostate stem cell antigen (PSCA), which are predominantly expressed in prostate tissue [29]. Numerous past and ongoing attempts have been made to induce immune responses, targeting these antigens. Sipuleucel-T is designed to invoke a T cell response to PAP [30,31], while PROSTVAC-VF is a recombinant vaccinia virus encoding human PSA [32]. Several other approaches are currently in clinical trials [33].

Despite the low overall response, there appears to be a subset of patients who have a sustained response to CPIs. These patients seem to have a greater intratumoral cluster of differentiation (CD)8 T cell density, high interferon gamma (IFNγ) response gene signature, and antigen-specific T cell responses [26]. There has been substantial interest in exploring biomarkers indicative of clinical benefit with the currently available prostate cancer treatments. For example, a persistent neutrophil to lymphocyte ratio greater than three during treatment with enzalutamide has been associated with a favorable clinical outcome [34]. The presence of androgen receptor splice variant-7 (AR-V7) expression is associated with resistance to enzalutamide and abiraterone [35]. Recently, a cell-free DNA-based method of detecting AR locus alterations, including the AR enhancer, by an assay-EnhanceAR-Seq, was strongly correlated with resistance to AR-directed therapy and worse survival [36]. Increasing knowledge of such correlates of immune response could hold the key to improving patients’ response to immunotherapy.

## 3. Vaccines

Bacillus Calmette-Guérin (BCG) vaccine was the first FDA-approved immunotherapy for the treatment of any solid tumor (bladder cancer) in 1990 [37,38]. While the FDA-approval of Sipuleucel-T followed it for mCRPC in 2010, unfortunately, since then, there have not been additional vaccine approvals for prostate cancer [39]. As of July 2020, there are thirteen vaccine clinical trials actively recruiting patients with prostate cancer. Of these, three are being evaluated in patients with mCRPC in combination with CPIs. While vaccine-based therapies have several advantages, one possible drawback is that an effective immune response to a specific TAA might be variable, limited in part by human leukocyte antigen (HLA) expression and haplotype, which affect presentation of the immunogenic epitope(s) [40,41]. High-affinity peptide-MHC (major histocompatibility complex) interactions and increased duration of peptide-MHC interactions may lead to more effective vaccine-induced immunogenicity [42,43]. Vaccine-based therapies can be broadly categorized into four different types of approaches as described below and schematically represented in Figure 1.

### 3.1. Deoxyribonucleic Acid (DNA)-Based Vaccines

DNA-based vaccines are commonly plasmids taken up by host cells, resulting in host-synthesized TAAs, which elicit an immune response (Figure 1A) [44,45]. As an example, pTVG-HP and pTVG-AR are plasmid DNA vaccines encoding the human PAP and AR antigens, respectively, and are currently being evaluated in clinical trials in mCRPC [46,47,48]. In addition to shared TAAs such as AR and PAP, the DNA vaccine platform can generate personalized cancer vaccines for prostate cancer patients [49], such as the ongoing phase I clinical trial that utilizes a combination of a neoantigen DNA vaccination, nivolumab, ipilimumab, and PROSTVAC for patients with mHSPC (NCT03532217), which takes advantage of both shared and personalized antigen approaches. However, whether or not a personalized approach represents an improvement over the utilization of shared tumor antigens remains to be seen.

### 3.2. Peptide-Based Vaccines

Peptide-based vaccines are built of subunits containing the specific epitope of an antigen (Figure 1D) [50]. UV1 is a synthetic long peptide vaccine containing a fragment of human telomerase reverse transcriptase (hTERT) administered in combination with granulocyte-macrophage colony-stimulating factor (GM-CSF) in a phase I/II clinical trial for patients with mHSPC (NCT01784913). hTERT plays a pivotal role in cancer development. It is responsible for the immortality and “stemness” of tumor cells and is often overexpressed in cancer cells [51]. Another rapidly evolving approach is developing personalized peptide vaccines that involve identifying an individual patient’s peptide candidates for their ability to induce an immune response in vitro and subsequent administration to the patient [52]. These have the potential to induce clinically meaningful, potent anti-tumor responses.

### 3.3. Viral Vector-Based Vaccines

This paradigm involves transferring a gene encoding TAA(s) into patients using a vector, in this case, a virus, resulting in stimulation of host immune response against the antigen (Figure 1C) [53,54]. PROSTVAC is a therapeutic vaccine that encodes PSA to generate a T-cell response. It utilizes two different live poxviral-based vectors-PROSTVAC-V, a recombinant vaccinia virus (rilimogene galvacirepvec), and PROSTVAC-F, a recombinant fowlpox virus (rilimogene glafolivec). In addition to induced modified human PSA, they contain three costimulatory domains for T cells, called TRICOM. These are B7.1, leukocyte function-associated antigen-3, and intercellular adhesion molecule-1 [55]. The initial phase I and II trials incorporated the use of granulocyte-macrophage colony-stimulating factor (GM-CSF), which is hypothesized to potentiate T-cell responses [56,57]. The phase III PROSPECT trial failed to show a significant difference in OS [58,59]. It was suggested that the FDA approvals of newer agents such as cabazitaxel, sipuleucel-T, abiraterone acetate, enzalutamide, and radium-223 during the trial period and follow up increased OS in the placebo group, and that the initial expected OS at the time of protocol design was a flawed assumption [19,60,61,62,63,64,65].

### 3.4. Cell-Based Vaccines

For this vaccine approach, whole cells, autologous or allogeneic, are used as antigen sources (Figure 1B) [66]. The polyvalent source of antigen can theoretically bypass antigen escape [67,68]. These are usually utilized in combination with GM-CSF to induce the growth and differentiation of dendritic cells involved in antigen presentation [67].

One whole tumor cell-based vaccine approach is GVAX-PCa. It consists of two allogeneic prostate cancer cell lines, LNCaP and PC3, modified to secrete GM-CSF. It has been tested in two phase III trials, VITAL-1 and VITAL-2, and failed to show clinical benefit [68,69,70]. There are attempts to improve the efficacy of GVAX-PCa by combining it with CPIs [71].

As mentioned, Sipuleucel-T, an autologous cell vaccine prepared from peripheral blood mononuclear cells (PBMCs), is currently approved for the treatment of patients with mCRPC [30]. The patients undergo leukapheresis, and the cells are exposed in vitro to PA2024—a fusion protein consisting of human GM-CSF and human PAP [30]. Notably, the patients with visceral metastases and symptomatic cancer pain were excluded from the clinical trials [19,72,73]. There were two interesting observations from these trials. Firstly, progression-free survival (PFS) was not prolonged [19]. Secondly, median survival improved significantly in patients with a lower PSA than patients with a higher PSA [74]. The phase III IMPACT trial demonstrated that Sipuleucel-T improved OS by 4.1 months in patients with mCRPC compared to placebo [19].

## 4. Immune Checkpoint Inhibitors

CPIs are immunomodulatory antibodies that aim to enhance the immune response to cancer. At this time, the most commonly administered CPIs target the immune checkpoints programmed death 1/ programmed death ligand 1 (PD-1/PD-L1) and Cytotoxic T lymphocyte Antigen 4 (CTLA-4) [75]. While blockade of these checkpoints has led to improved responses and changed the treatment paradigm in malignancies such as melanoma, non-small cell lung cancer (NSCLC), efficacy in prostate cancer remains modest [22,76,77]. There have been ongoing efforts to identify predictors of response to immunotherapy, such as PD-L1 expression, tumor mutational burden (TMB), and other mutations such as CDK12 [78,79,80,81]. One study of tumor tissue from patients with mCRPC revealed that 19% demonstrated high PD-1/PD-L1 immunoexpression [82]. PD-L1 expression is upregulated in tumors with a higher T stage and lymph node involvement [83]. These patients represent a subset that benefits from treatment with CPIs, as demonstrated in the CheckMate 650 clinical trial [84]. Interestingly, standard of care hormone-based treatments used in prostate cancer may have differential effects on PD-L1 expression [85]. While enzalutamide has shown upregulation of PD-L1 in preclinical models, neoadjuvant treatment with abiraterone plus prednisone revealed a trend towards decreased PD-L1 positivity [86]. Below we review some of the CPI trials in prostate cancer to date.

### 4.1. CTLA-4 Inhibitor-Ipilimumab

CTLA-4 is a major inhibitory pathway for T lymphocytes [87]. Ipilimumab, a humanized anti-CTLA-4 monoclonal antibody, binds to and inhibits the CTLA-4 receptor on T lymphocytes, which leads to an enhanced anti-tumor immune response [88]. Two-phase II clinical trials (NCT00861614, NCT01057810) have failed to show a significant OS benefit of ipilimumab in patients with mCRPC [89,90]. Subsequent analyses have revealed that the presence of high intratumoral CD8 T cell density and IFN-γ response gene signature may be predictive of response to ipilimumab [26]. A slightly higher disease control rate was observed in patients with bone-predominant disease. Importantly, however, the final analysis of the phase 3 trial CA184-043, which evaluated radiotherapy to bone metastases followed by ipilimumab versus placebo in men with mCRPC, revealed the OS was two to three times higher for patients on ipilimumab at year 3 and beyond [91]. This finding suggests that while the short-term benefit of single-agent ipilimumab was limited, the establishment of long-term anti-tumor immunity appears to be clinically relevant.

### 4.2. PD-1 inhibitors-Nivolumab and Pembrolizumab

PD-1 and its ligands-PD-L1 and programmed death ligand 2 (PD-L2), like CTLA-4, deliver inhibitory signals for T cell activation [92]. Based on the results of phase Ib KEYNOTE-028 and phase II KEYNOTE-199 clinical trials, pembrolizumab has an insignificant response as monotherapy in patients with mCRPC [21,93]. In the KEYNOTE-199 study, the objective response rate (ORR) was 5% in patients with PD-L1 positive patients, defined as having a combined positive score (CPS) of ≥1 using the PD-L1 IHC 22C3 pharmDx assay, compared to 3% for patients with a negative PD-L1 expression [21].

It has been hypothesized that patients with a deficient mismatch repair (dMMR) mechanism express a greater number of neoantigens, leading to a more robust immune response with the administration of CPIs [94]. Pembrolizumab has received tissue agnostic approval to treat patients with dMMR who do not have an appropriate alternative treatment [95]. The response to pembrolizumab was also observed in another phase II trial, where pembrolizumab was administered to patients who progressed on enzalutamide [96]. At the time of publication, the study had enrolled ten patients, and there were two exceptional responders. These two patients underwent genetic analysis, and one of them exhibited microsatellite instability (MSI), which is a marker of dMMR.

The combination of ipilimumab and nivolumab has demonstrated some efficacy in AR-V7 expressing metastatic prostate cancer [97]. The ORR in the phase II study was 25% in patients with measurable disease, and a trend towards increased ORR was noted in patients with DNA-repair deficiency (DRD) [97]. The response might be because AR-V7 mutations are associated with an increase in mutations in the DNA repair pathway and a higher TMB [98,99]. The recently published CheckMate 650 clinical trial enrolled 90 patients with mCRPC, suggested an improved response to CPIs in tumors expressing a relatively high TMB [84]. The ORR in the pre-chemotherapy cohort was 25%, versus 10% in the post-chemotherapy cohort.

Numerous PD-1 inhibitors are currently undergoing clinical trials in patients with mCRPC. Cemiplimab is a PD-1 inhibitor approved for the treatment of patients with cutaneous squamous cell carcinoma (CSCC) [100]. Cemiplimab is currently undergoing clinical trials combined with other agents for patients with prostate cancer, such as the novel mRNA vaccine W_pro1 (NCT04382898). W-pro1 is a messenger ribonucleic acid (mRNA) vaccine complexed with liposomes and encodes five antigens expressed in de novo and metastatic prostate cancer.

## 5. Potential Immune Evasion Mechanisms

The TME comprises immune cells, non-immune cells such as fibroblasts, and endothelial cells embedded in an extracellular matrix (ECM) [101]. The complex interplay between protumorigenic and anti-tumor immune-modulating factors leads to the varying patterns of tumor progression, and our therapeutic decisions influence these opposing factors. The immune evasion mechanism can be categorized into factors that increase or dampen the immune response and are briefly outlined below and depicted schematically in Figure 2.

### 5.1. Increase Immune Suppressive Factors

There are a variety of immune-suppressive factors in the TME, such as PD-L1, interleukin 10 (IL-10), hypoxia-inducible factor 1-alpha (HIF-1α), and transforming growth factor-β (TGF-β) [102,103,104,105]. There is evidence suggesting an increase in myeloid-derived cells, tumor-associated macrophages (TAMs), and myeloid-derived suppressor cells (MDSCs) in the TME of prostate cancer is associated with tumor progression [106]. The majority of the studies show that pathways such as the loss of phosphatase and tensin homolog (PTEN) increase MDSC infiltration [107]. Chronic inflammation induced by the tumor, inflammatory cytokines such as soluble tumor necrosis factor (sTNF), interleukin 1 beta (IL-1β), TGF-β, and IL-10 cause myeloid cells to differentiate into MDSCs, which have been implicated in worsened prognosis, and resistance to CPI immunotherapy [108,109,110].

TAMs express reduced major histocompatibility complex (MHC) expression and lead to an increase in PD-L1 expression [111]. Increased regulatory T cells (Tregs) in prostate cancer specimens are associated with advanced stage and worsened prognosis [112]. There is an increase in the level of TGF-β in patients with prostate cancer [113], which leads to an increase in tumor growth [114]. Mutations in forkhead box protein A1 (FOXA1), a transcription factor essential for epithelial lineage differentiation, induces TGF-β signaling in CRPC [115].

### 5.2. Reduced Immune Stimulatory Factors

There is evidence that peripheral natural killer (NK) cell pools may be decreased in patients with mCRPC. One potential mechanism for this decrease in peripheral NK cell pool is an increased NK cell group 2 member D (NKG2D) serum receptor levels derived from the tumor [116]. Membrane-bound and not tumor shed NKG2D ligand is thought to promote CD8 T cell and NK cell response [117]. Circulating NK cell function is reduced in patients with prostate cancer, and this difference is more profound in patients with metastatic disease [106,116]. Moreover, ADT is associated with an increase in CD8 T cells [106]. The incidence of the T-cell receptor zeta (TCR ζ) chain has been correlated with adequate effector cell function, and flow cytometric analysis has revealed a reduction in ζ chain expression in peripheral blood lymphocytes (PBL) of patients with prostate cancer [118].

Tumors with a low TMB, such as prostate cancer, often have reduced expression of neoantigens, which can be recognized by the immune system [119]. Tumors with higher T cell trafficking have been shown to have a higher therapeutic response to immune therapy [120]. Malignant transformation is often associated with changes in HLA expression [121,122,123], and thus these defects in HLA expression and antigen presentation machinery can lead to decreased tumor cell recognition by cognate T cells. This can result in immune evasion by cancer cells and may have therapeutic implications with respect to response to CPIs [124]. This also provides additional rationale for the development of HLA-independent immuno-cell therapies [125].

The impact of therapy on immune responses is complex. ADT can lead to enhanced lymphopoiesis and mitigate immune tolerance to prostate cancer antigens [126]. Androgen receptor (AR) antagonists have been shown to inhibit T cell responses, possibly related to γ-Aminobutyric acid type A (GABA-A) off-target effects [127]. It is postulated that initiating immunotherapy before AR antagonist therapy could improve tumor control by upregulation of the immune response [128]. As our understanding of tumor biology and the effect of treatments evolve, it could influence the choice and sequence of therapies in the future.

## 6. Ongoing Clinical Trials on Combination with CPIs

Below we discuss some of the ongoing attempts to improve the response to existing immunotherapeutic agents and novel immunotherapeutic agents currently in clinical trials (Figure 3A and Figure 4B). A breakdown of the currently ongoing combination treatments of CPIs with other agents is schematically depicted in Figure 3B,C.

### 6.1. Vaccines with CPIs

As of July 2020, three ongoing trials are actively recruiting patients with mCRPC, which combine vaccines with CPIs (Figure 4C). One potential limitation of using self-antigen TAAs as therapeutic vaccine targets is the issue of central T cell tolerance, which must be overcome in order to mount an effective anti-tumor response [129]. CPIs, when combined with vaccines encoding specific TAAs, have the potential to expand antigen-specific T cells [130]. Tumor antigens used in therapeutic vaccines are processed and presented in the context of MHC, and thus predicting these peptide: MHC binding interactions using advanced algorithms to improve T cell responses is of intense clinical interest [131,132]. Thus, there is a sound scientific rationale for combining vaccines with CPIs to potentiate the vaccine’s immune response [133,134,135].

One of the vaccines, ChAdOx1-MVA 5T4, is a combination of two replication-deficient viruses, chimpanzee adenovirus and modified vaccinia ankara, targeting an oncofetal self-antigen 5T4 (NCT03815942) [136]. The 5T4 protein is expressed in numerous malignancies but seldom in normal tissue [137]. It is associated with cells’ movement through the epithelial-mesenchymal transition [138], modulation of chemokine, and Wnt signaling [139,140]. A phase I study revealed 5T4-specific T cell responses [141].

Another vaccine that is currently recruiting patients in a phase II trial is the combination of two DNA vaccines, pTVG-HP and pTVG-AR, which target PAP and the androgen receptor ligand-binding domain (AR LBD), respectively, in combination with Pembrolizumab (NCT04090528) (Figure 4E) [142,143]. pTVG-HP monotherapy, in a phase II trial, revealed a possible effect on micrometastatic bone disease [48]. In a phase I study, patients who received pTVG-AR developed a T helper type 1 (Th1) biased immunity to the AR LBD [47].

The third vaccine in clinical trials is Wpro1 (NCT04382898). It is an mRNA-based vaccine that encodes for five antigens expressed in prostate cancer. Practical applications of mRNA-based vaccines are relatively recent, as they have been limited by instability and inefficient in vivo delivery [144]. Technological advances in delivery platforms have overcome these challenges [145], and several mRNA vaccines are currently in clinical trials. The mRNA is complexed with liposomes to form serum-stable RNA lipoplexes and induce activation of vaccine antigen-specific CD8 and CD4 T cells, in addition to the innate immune system.

The development of genomic approaches that allow the identification of tumor neoantigens has led to the development of clinical trials of personalized neoantigen vaccines [146]. Emerging clinical data reveals that targeting these mutated neoantigens can result in tumor responses [147,148]. This approach will likely lead to the development of clinical trials for patients with mCRPC in the future.

### 6.2. Combinations of CPIs and TKIs

Due to CPI monotherapy’s limited activity in prostate cancer [20,21], there has been enthusiasm for combining these agents with tyrosine kinase inhibitors (TKIs) [149]. TKIs compete with the adenosine triphosphate (ATP) binding site of the catalytic domain of oncogenic tyrosine kinases [150]. TKIs such as cabozantinib, sunitinib, and axitinib can act through several immunomodulatory mechanisms, such as anti-angiogenesis by inhibition of vascular endothelial growth factor (VEGF) [151]. VEGF overexpression is associated with the inhibition of differentiation of monocytes into dendritic cells [152]. Inhibition of VEGF leads to reduced immune inhibitory stimuli such as Tregs and MDSCs [153,154]. Cabozantinib has also been shown to target tyrosine-protein kinases-c-Met, Tyro3, Axl, and Mer, which have a role in immunosuppression [155,156].

The combination of TKI and CPI has already shown synergy and has been approved in NSCLC and renal cell carcinoma (RCC) [157,158]. Cabozantinib was evaluated in combination with PD-L1 inhibitor atezolizumab, in the phase 1b clinical trial COSMIC-021. One out of the 44 patients enrolled in the mCRPC cohort experienced a grade 5 treatment-related adverse event (TRAE) of dehydration. The ORR was 32%, while 48% of the patients had stable disease, leading to a disease control rate of 80% [159]. The combination of cabozantinib and atezolizumab for patients with mCRPC is also being evaluated in a phase III clinical trial (NCT04446117).

### 6.3. Combinations of CPIs and Chemotherapy

While chemotherapies are typically immune-suppressive, they can potentiate anti-tumor immunity by enhancing antigen presentation, expressing costimulatory molecules, and downregulating PD-L1 expression [160,161]. This potential synergy has led to the approval of the combination of CPI and chemotherapy in patients with NSCLC and is considered the front line in the metastatic setting [162]. There are currently six ongoing clinical trials that recruit patients with mCRPC, which evaluate a CPI’s efficacy in combination with chemotherapy (Figure 4K). KEYNOTE-365 and KEYNOTE-921 phase I and III trials, respectively, evaluate the combination of PD-1 inhibitor pembrolizumab with docetaxel (NCT02861573, NCT03834506). CheckMate 9KD evaluates the PD-1 inhibitor nivolumab in combination with docetaxel (NCT03338790). PT-112 is a novel molecule comprised of a platinum agent complexed to a pyrophosphate ligand [163]. It has potential synergy with CPIs and is currently being evaluated in combination with the PD-L1 inhibitor avelumab in the phase I/II PAVE-1 clinical trial (NCT03409458).

### 6.4. Combination of CPIs with Radiopharmaceuticals

There is evidence that targeted radionuclide therapy (TRT) may increase PD-L1 expression on T cells and that the combination of TRT with CPI leads to increased infiltration by CD8 T cells [164]. As of July 2020, there are 20 interventional clinical trials actively recruiting patients that test radiopharmaceuticals’ efficacy in patients with mCRPC. A phase Ib and a phase I/II trial combine pembrolizumab with Lu 177-PSMA-617, a β emitting isotope of lutetium attached to PSMA (NCT03658447, NCT03805594) (Figure 4I). Lu 177-PSMA-617 has shown some efficacy as monotherapy in mCRPC [165]. The combination of radium-223, PD-L1 inhibitor avelumab, and peposertib is being evaluated in a multicenter phase 1/2 clinical trial (NCT04071236). Peposertib is a DNA-dependent protein kinase inhibitor, thereby inhibiting its ability to function in DNA damage response (DDR). By preventing the repair of radiation-induced DNA double-strand breaks (DSBs), peposertib has a potential for synergy with radiation [166].

### 6.5. Combination of CPIs with Radiation

Stereotactic body radiation therapy (SBRT) offers the benefit of local control for oligometastatic disease with minimal toxicity [167]. There is some evidence that suggests treatment with radiation leads to antigen release and exposure of damage-associated molecular patterns—potentially leading to the “abscopal effect” [168]. In addition to upregulation of MHC I expression, which is potentially immune-stimulating when combined with CPI [169,170], radiation has other potential synergistic effects with CPI, such as increased Fas surface expression leading to interaction with Fas ligand (FAS-L) and enhanced cytotoxic T cell activity [171], dendritic cell activation resulting in enhanced cross-presentation of TAAs [172], and upregulation of PD-L1 [173,174].

Post-hoc analysis of the KEYNOTE-001 trial suggests previous treatment with radiation in patients with NSCLC leads to improved PFS and OS with pembrolizumab therapy [175]. The efficacy of PD-L1 inhibitor durvalumab after concurrent chemoradiotherapy for patients with NSCLC was demonstrated in the phase III PACIFIC clinical trial [176]. The phase I RADVAX clinical trial evaluating the combination of pembrolizumab with hypofractionated radiotherapy (HFRT) in patients with metastatic cancers revealed that in some patients, HFRT could reinvigorate systemic response to pembrolizumab, despite prior progression on PD-1 therapy [177]. One of the PORTER trials’ cohorts evaluates the combination of nivolumab with SBRT in patients with mCRPC (NCT03835533).

### 6.6. Combination of CPIs with PARP Inhibitors

Poly ADP ribose polymerase (PARP) inhibitors are small molecules that block the repair of single-strand DNA breaks. Because of this, tumors with homologous recombination deficiency (HRD) show particular sensitivity to PARP inhibition [178]. PARP inhibitors have multiple potential synergistic effects when combined with CPIs, such as increased intratumoral CD8 T cell infiltration [179], increased IFNγ production in the TME [180], and PD-L1 upregulation [181].

There are currently two PARP inhibitors approved for administration in patients with mCRPC and HRD. Rucaparib was approved based on the phase II TRITON2 study’s preliminary results, which evaluated men with HRD CRPC [182]. Rucaparib demonstrated an ORR of 44% in patients with BRCA 1/2 mutation and is currently approved for this patient population. Olaparib was approved based on the results of the phase III PROFOUND clinical trial, which enrolled patients with mCRPC who had disease progression while receiving a new hormonal agent (e.g., enzalutamide or abiraterone), and had a qualifying alteration in prespecified genes involved in homologous recombination repair [183]. Olaparib demonstrated a higher ORR in men with previously treated mCRPC and HRD compared to the physician’s choice of enzalutamide or abiraterone (control), and is FDA approved for this patient population.

PARP inhibitors could have efficacy beyond HRD tumors when combined with other agents such as cytotoxic chemotherapy (taxanes, topoisomerase inhibitors, etc.) and radiation due to the accumulation of DNA damage [184]. As of July 2020, there are 14 ongoing trials testing this hypothesis. These ongoing clinical trials evaluate PARP inhibitors’ efficacy in combination with various agents such as chemotherapy, radiopharmaceuticals, and checkpoint inhibitors (Figure 4J). Of these, only five clinical trials are in patients with HRD. PARP inhibition could potentiate immune modulation elicited by CPI by enhanced immune priming and T cell infiltration [185]. KEYNOTE-365 is evaluating the combination of pembrolizumab and olaparib (NCT02861573). Similarly, CheckMate 9KD evaluates the combination of nivolumab and rucaparib [186].

### 6.7. Combinations of CPIs and Adenosine Receptor Antagonists

A2B adenosine receptor (A2BR) is a G protein-coupled receptor (GPCR), which is expressed at high levels in prostate cancer tissue [187], and its blockade has been associated with inhibition of prostate cancer growth [188]. It has also been observed that PD-1 blockade leads to overexpression of adenosine 2A receptor (A2AR) on tumor-infiltrating CD8 T cells, thereby making them more susceptible to A2A-mediated suppression [189]. Currently, five ongoing trials are evaluating the efficacy of adenosine receptor antagonists in patients with mCRPC. Of these, four combine adenosine receptor antagonists with CPI (NCT04381832, NCT03207867, NCT04089553, NCT03629756) (Figure 4G). NZV930 is a novel anti-CD73 monoclonal antibody that prevents the conversion of extracellular adenosine monophosphate (AMP) to adenosine and is being evaluated in a phase Ib/II study in combination with a CPI (NCT03549000).

### 6.8. Combination of Interleukin-2 (IL-2) Agonist with CPI

IL-2 is a cytokine that potentiates T cell proliferation and NK cell activity and induces differentiation of Tregs [190]. Based on the interim results of the phase I/II PIVOT-2 clinical trial, IL-2 receptor agonist bempegaldesleukin has demonstrated activity in combination with nivolumab for patients with urothelial carcinoma (Figure 4A). Notably, it converted some PD-L1 negative patients to PD-L1 positive, as assessed by the PharmDx 28-8 assay [191]. It is currently being evaluated in a phase I/II clinical trial in combination with avelumab (NCT04052204).

### 6.9. Combination of CD11b Agonist with CPI

CD11b is an integrin that promotes pro-inflammatory macrophage polarization and suppresses tumor growth [192]. Preclinical studies have shown it can lead to a reduction in MDSCs and enhance dendritic cell responses, all of which potentially improve anti-tumor T cell immunity and may have synergistic effects with CPIs [193,194]. GB1275 is a CD11b agonist being tested in a phase I/II clinical trial combined with pembrolizumab in patients with solid cancers, including mCRPC (NCT04060342) (Figure 4D).

## 7. Additional Ongoing Strategies to Overcome Immune Resistance

Multiple strategies are currently being tested to overcome immune resistance, some of which are depicted schematically in Figure 4. A few strategies with the potential to have a major impact on prostate cancer therapy are summarized below.

### 7.1. Cellular Therapies

Chimeric antigen receptor (CAR) T cells are autologous T cells engineered to express a synthetic receptor with a high affinity for tumor cells [195]. They can target TAAs in a non-HLA complex-restricted manner [196]. Bi-specific T-cell engager antibody constructs genetically link a polypeptide chain, usually a TAA’s minimal binding domain, to a T-cell receptor, such as CD3 or CD28, leading to T cell activation and resulting in tumor cell lysis [197]. There are currently two approved CAR T cells directed against CD19 for the treatment of acute lymphoblastic leukemia (ALL) and diffuse large B-cell lymphoma (DLBCL) [198,199]. While cellular therapies such as blinatumomab, a CD3/CD19 Bi-specific T-cell engager [200], and tisagenlecleucel, an anti-CD19 CAR T cell therapy in ALL [198], and axicabtagene ciloleucel, another anti-CD19 CAR T cell therapy in DLBCL, have shown remarkable efficacy [199], this success has not yet been replicated in solid malignancies [201]. This lack of efficacy is likely due to the immunosuppressive TME, reduced trafficking to tumor sites, tumor heterogeneity, and difficulty for T cells to infiltrate the tumor [202,203,204].

Four clinical trials are currently evaluating CAR T cells that target the TAA PSMA (NCT04227275, NCT04053062, NCT04249947, NCT03089203), and two that evaluate ones that target the TAA PSCA in patients with mCRPC (NCT03873805, NCT02744287). In addition, there are three PSMA × CD3 (NCT04104607, NCT03792841, NCT04104607), one PSMA × CD28 (NCT03972657), one six transmembrane epithelial antigen of the prostate 1 (STEAP1) × CD3 (NCT04221542), and an anti-Her2 bispecific antibody (Her2Bi) × CD3 (NCT03406858) bi-specific T-cell engager clinical trials ongoing (Figure 4H). One strategy to enhance the activity of bi-specific T-cell engagers in prostate cancer is to administer it with a CPI. A few of the trials mentioned above are testing this concept’s feasibility and efficacy [205].

### 7.2. Alternative Immune Checkpoints

Utomilumab is a monoclonal agonist antibody that binds to the 4-1BB (CD-137) receptor on T cells and NK cells, leading to increased immune cell infiltration (Figure 4F) [206]. OX40 is a receptor found on both activated CD4 and CD8 T cells, in addition to other lymphoid and non-lymphoid cells, but not on resting naïve T cells [207]. PF-04518600 is an OX40 receptor agonistic monoclonal antibody, leading to increased infiltration of T cells and inflammatory cytokines and reduced infiltration of immune-suppressive Tregs [208]. The combination of utomilumab, PF-04518600, and avelumab is being evaluated in a phase I/II clinical trial for patients with mCRPC (NCT03217747).

B7-H3 is an inhibitory immune checkpoint that is overexpressed in many advanced prostate cancers [209]. Enoblituzumab is an inhibitory monoclonal antibody against B7-H3, currently being evaluated in localized prostate cancer [210] and has potential for future applications in mCRPC.

## 8. Conclusions

While effective CPI strategies leading to an OS benefit in an unselected advanced prostate cancer population have not yet been identified, there were signs of increased OS with ipilimumab at later time points (NCT01057810) [89]. The demonstrated benefit of pembrolizumab in patients with dMMR is a notable exception [211]. In addition, except for sipuleucel-T, monotherapy with vaccines or CPIs in prostate cancer patients has been mostly unsuccessful.

As our understanding of tumor biology increases, we are beginning to exploit alternative pathways to increase immunotherapy response in patients with mCRPC. The current treatment paradigm is to design clinical trials that combine agents with complementary anti-tumor activity, with an overarching goal of transforming an immunosuppressive “cold tumor” to an immunotherapy responsive “hot tumor”. The novel combinatorial approaches summarized above could lead to synergy and thus enhanced responses in patients. With technological advances in genomics, personalized vaccines are currently being tested in prostate cancer (NCT03532217) and will likely be utilized with increasing frequency in future clinical trials.

Despite the lack of past successes, the future for immunotherapy in prostate cancer is optimistic. Scientists’ and clinicians’ persistence in enrolling appropriate patients in carefully designed clinical trials holds the key to future successes.

## Figures and Tables

**Figure 1 cancers-13-00334-f001:**
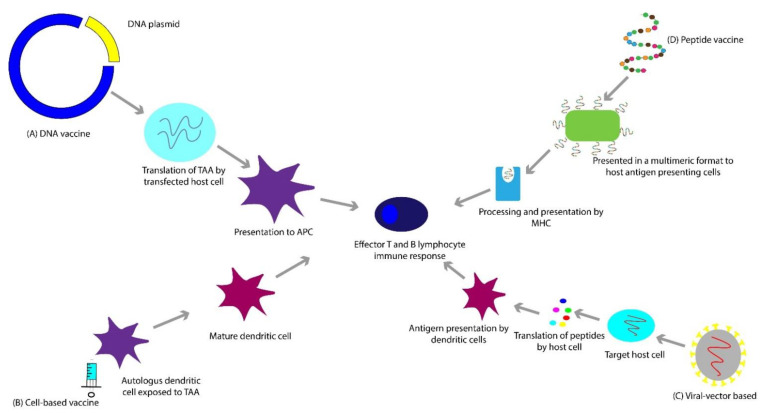
(**A**) DNA vaccines employ a plasmid, which gets translated to a tumor-associated antigen. (**B**) Cell-based vaccines utilize whole cells as a source of antigen. (**C**) Viral-vector based vaccines are based on the process of transfecting the host cell with a viral vector, that contains a gene which encodes a tumor-associated antigen. (**D**) Peptide-based vaccines consist of subunits of an epitope of a tumor-associated antigen, sometimes presented in a multimeric format (e.g., virus-like particles (VLPs)) to elicit an effective immune response.

**Figure 2 cancers-13-00334-f002:**
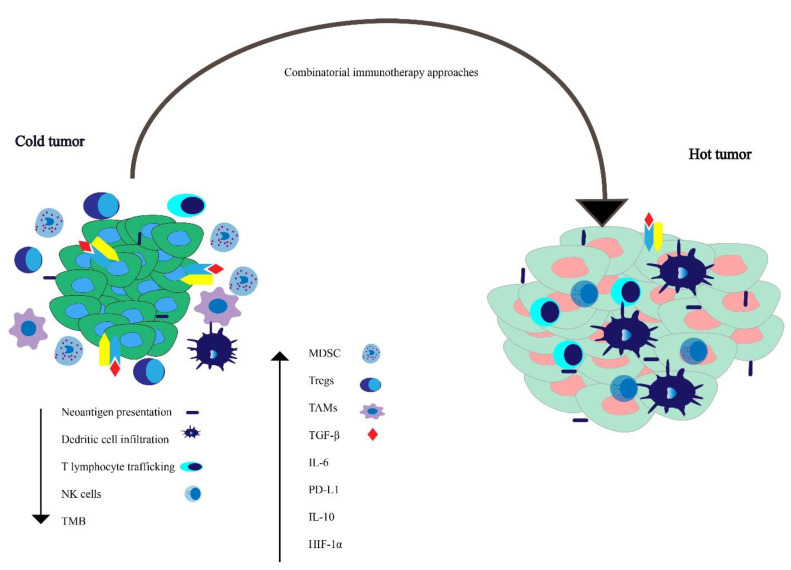
Working model for understanding factors influencing tumor microenvironment in tumors lacking immunological anti-tumor activity—the so-called “cold tumors”—versus tumors with immunological activity against the tumor—“hot tumors.” Combinatorial immunotherapeutic approaches can possibly convert an immunologically cold tumor to a hot tumor.

**Figure 3 cancers-13-00334-f003:**
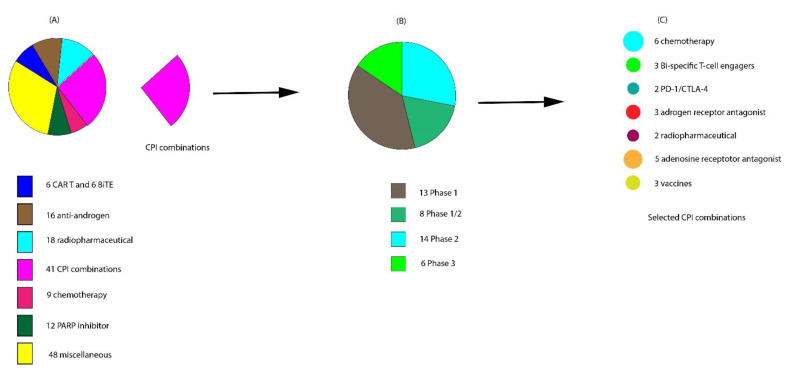
Snapshot of ongoing interventional clinical trials actively recruiting patients with metastatic castration-resistant prostate cancer (mCRPC) in July 2020. (**A**) Breakdown of the 150 clinical trials ongoing. (**B**) Of the 41 checkpoint inhibitor (CPI) trials, 13 are in phase I, 8 in phase I/II, 14 in phase II, and 6 in phase III. (**C**) The agents currently being used for CPI combinatorial therapy.

**Figure 4 cancers-13-00334-f004:**
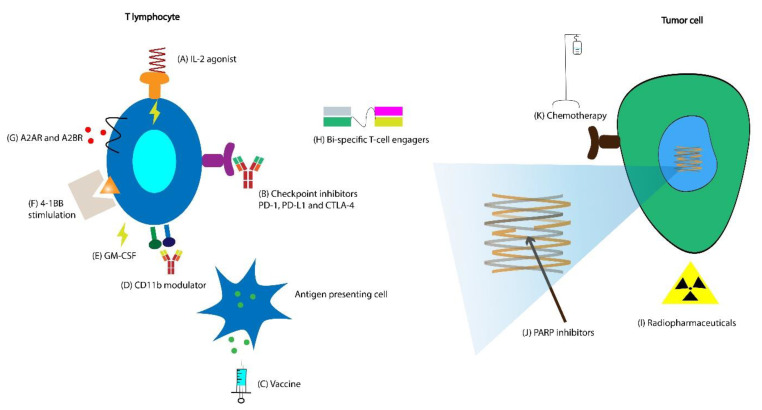
(**A**) Bempegaldesleukin is an interleukin (IL)-2 agonist currently being tested in combination with avelumab. (**B**) Multiple new checkpoint inhibitors (CPIs) are in clinical trials in combination with other agents. (**C**) Currently, there are DNA-, viral vector-, and mRNA-based vaccines in combination with checkpoint inhibitors in clinical trials in mCRPC. (**D**) GB1275 is a CD11b agonist in clinical trials in combination with CPI. (**E**) Granulocyte-macrophage colony-stimulating factor (GM-CSF) is used in combination with vaccines to augment immunological response (NCT04090528). (**F**) 4-1BB stimulation has been employed extensively in chimeric antigen receptor (CAR) T cell therapies, and utomilumab, a monoclonal agonist antibody that binds to the 4-1BB is currently in clinical trials for patients with mCRPC. (**G**) Currently, there are four clinical trials combining adenosine receptor antagonists with checkpoint inhibitors in patients with mCRPC. (**H**) After their success in acute lymphoblastic leukemia (ALL) and diffuse large B-cell leukemia (DLBCL), cellular therapies such as CAR T cell and bi-specific T-cell engagers are being studied extensively in clinical trials in numerous solid malignancies, including prostate cancer. (**I**) There are two clinical trials combining checkpoint inhibitor therapy with radiopharmaceutical Lu 177-PSMA-617, which is a β emitting isotope of lutetium attached to prostate-specific membrane antigen (PSMA). (**J**) While poly ADP ribose polymerase (PARP) inhibitors have already demonstrated efficacy in patients with homologous recombination deficiency (HRD), clinical trials are currently ongoing in combination with agents such as radiopharmaceuticals, CPIs, and chemotherapy evaluating whether they are effective in patients without HRD. (**K**) Chemotherapeutic agents have potential synergy with checkpoint inhibitors, and clinical trials to test this hypothesis in patients with mCRPC are currently ongoing.

## Data Availability

No new data were created or analyzed in this study. Data sharing is not applicable to this article.

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
