# Peer review of "Immunotherapy and Immunotherapy Combinations in Metastatic Castration-Resistant Prostate Cancer"

_cancers, 2021, doi:10.3390/cancers13020334_

Round 1

Reviewer 1 Report

In this review manuscript Dr Bansal and coauthors provide a complete and instructive overview of the main current immunotherapy strategies, with relative ongoing clinical trials, against castration-resistant prostate cancer.

The manuscript is well written. It is very informative from the clinical perspective while it not always reaches the necessary depth about the immunological mechanisms underlying the described strategies.   

It would be helpful if, beside the useful list of ongoing clinical trials, the authors could add some comments and expert opinion on the potential advantages or risks that each strategy might bring.  Just as an example (but the above comment may be extended to all manuscript’s sections), it could be important to underscore that vaccines based on a precise TAA would be limited to patients with a given HLA haplotype capable to efficiently “restrict and present” the immunogenic epitope.

- It could be appropriate to add just few lines of rationale sustaining the association of vaccines with checkpoint inhibitors, similarly to what is reported in the chapters dedicated to Chemo or TKi combinations.

 - Furthermore, defects in HLA expression and antigen processing machinery may be included among the immune-evasion mechanisms.  

- Line 82, please correct, “approved” is repeated twice.

- Reference 77 is incomplete

Author Response

Response to Reviewer 1 Comments

Point 1: The manuscript is well written. It is very informative from the clinical perspective while it not always reaches the necessary depth about the immunological mechanisms underlying the described strategies.  

It would be helpful if, beside the useful list of ongoing clinical trials, the authors could add some comments and expert opinion on the potential advantages or risks that each strategy might bring.  Just as an example (but the above comment may be extended to all manuscript’s sections), it could be important to underscore that vaccines based on a precise TAA would be limited to patients with a given HLA haplotype capable to efficiently “restrict and present” the immunogenic epitope.

Response 1: This is a very insightful observation and critical information previously lacking from this manuscript. We thank the reviewer for bringing this to our attention. We have incorporated the information where we thought it lacked throughout the manuscript. We specifically bring your attention to line numbers 93-97, 122-127, 260-265, 363, 380-382, 422-424, 433.

Point 2: It could be appropriate to add just few lines of rationale sustaining the association of vaccines with checkpoint inhibitors, similarly to what is reported in the chapters dedicated to Chemo or TKi combinations.

Response 2: We appreciate the reviewer pointing this out, as it adds to the manuscript's depth of information. Please see line numbers 281-287 for the addition.

Point 3: Furthermore, defects in HLA expression and antigen processing machinery may be included among the immune-evasion mechanisms. 

Response 3: We agree with the reviewer’s astute point that this is an important mechanism that should’ve been mentioned in the section on immune-evasion mechanisms. Please see line numbers 260-265 for the relevant addition.

Point 4: Line 82, please correct, “approved” is repeated twice.

Response 4: We appreciate the reviewer's close attention to our manuscript and have corrected this error. Please see line 88 in the updated manuscript for the correction.

Point 5: Reference 77 is incomplete

Response 5: We again appreciate the reviewer’s detailed reading of our manuscript and have updated the references to match the MDPI format.

Please note that all the referenced line numbers are relevant to the “simple markup” setting on Microsoft word.

Reviewer 2 Report

I read with very interest your manuscript. You often repeat the same words.  In Introduction and especially in the paragraph of Vaccines, some peroiods are convoluted

Author Response

Response to Reviewer 2 Comments

Point 1: I read with very interest your manuscript. You often repeat the same words.  In Introduction and especially in the paragraph of Vaccines, some periods are convoluted

Response 1: We thank the reviewer for their interest in our manuscript and for pointing out the use of redundant words throughout the manuscript. While the whole manuscript has been edited for style, we bring your attention specifically to line numbers 43-36, 50-53, 88-91 for some relevant changes we have made. We feel that these and other changes that the reviewer was kind enough to bring to out attention have substantially improved the manuscript.

Please note that all the referenced line numbers are relevant to the “simple markup” setting on Microsoft word.

Reviewer 3 Report

The manuscript titled "Immunotherapy and Immunotherapy Combinations in Metastatic Castration-Resistant Prostate Cancer", by Dhruv Bansal et al. is quite attractive; the issue is very interesting.

The manuscript needs of some revisions:

  1. The discussion needs to be implementate with some points about the role of pd1 and pdl1 in prostate cancer: please cite for example the following paper: Massari F, et al. MAGNITUDE OF PD-1, PD-L1 AND T LYMPHOCYTE EXPRESSION ON TISSUE FROM CASTRATION-RESISTANT PROSTATE ADENOCARCINOMA: AN EXPLORATORY ANALYSIS. Target Oncol. 2016 Jun;11(3):345-51.
  2. The discussion needs to be implementate with the role of some markers (clinical or biomarker? AR-V7; N/L ratio); please cite for example the following papers:
    1. Conteduca V, et al. PERSISTENT NEUTROPHIL TO LYMPHOCYTE RATIO >3 DURING TREATMENT WITH ENZALUTAMIDE AND CLINICAL OUTCOME IN PATIENTS WITH CASTRATION-RESISTANT PROSTATE CANCER. PLoS One. 2016 Jul 19;11(7):e0158952.

Author Response

Response to Reviewer 3 Comments

Point 1: The discussion needs to be implementate with some points about the role of pd1 and pdl1 in prostate cancer: please cite for example the following paper: Massari F, et al. MAGNITUDE OF PD-1, PD-L1 AND T LYMPHOCYTE EXPRESSION ON TISSUE FROM CASTRATION-RESISTANT PROSTATE ADENOCARCINOMA: AN EXPLORATORY ANALYSIS. Target Oncol. 2016 Jun;11(3):345-51

Response 1: We thank the reviewer for bringing our attention to this critical information missing from the manuscript. This information has now been added to the manuscript. Please see line numbers 167-175 and reference 85 for the relevant additions.

Point 2: The discussion needs to be implementate with the role of some markers (clinical or biomarker? AR-V7; N/L ratio); please cite for example the following papers:

Conteduca V, et al. PERSISTENT NEUTROPHIL TO LYMPHOCYTE RATIO >3 DURING TREATMENT WITH ENZALUTAMIDE AND CLINICAL OUTCOME IN PATIENTS WITH CASTRATION-RESISTANT PROSTATE CANCER. PLoS One. 2016 Jul 19;11(7):e0158952.

Response 2: We thank the reviewer for this astute observation. The relevant discussion has been added in line numbers 78-86 and reference 34.

Please note that all the referenced line numbers are relevant to the “simple markup” setting on Microsoft word. Also, we have reviewed the complete manuscript and edited for spelling errors and stylistic errors.

Round 2

Reviewer 3 Report

The manuscript is acceptable in this reviewed form.